# BIAS-VARIANCE DECOMPOSITION FOR BOLTZMANN MACHINES

## ABSTRACT

We achieve bias-variance decomposition for Boltzmann machines using an information geometric formulation. Our decomposition leads to an interesting phenomenon that the variance does not necessarily increase when more parameters are included in Boltzmann machines, while the bias always decreases. Our result gives a theoretical evidence of the generalization ability of deep learning architectures because it provides the possibility of increasing the representation power with avoiding the variance inflation.

## 1 INTRODUCTION

Understanding why the deep learning architectures can generalize well despite their high representation power with a large number of parameters is one of crucial problems in theoretical deep learning analysis, and there are a number of attempts to solve the problem with focusing on several aspects such as sharpness and robustness (Dinh et al., 2017; Wu et al., 2017; Keskar et al., 2017; Neyshabur et al., 2017; Kawaguchi et al., 2017). However, the complete understanding of this phenomenon is not achieved yet due to the complex structure of deep learning architectures.

To theoretically analyze the generalizability of the architectures, in this paper, we focus on Boltzmann machines (Ackley et al., 1985) and its generalization including higher-order Boltzmann machines (Sejnowski, 1986; Min et al., 2014), the fundamental probabilistic model of deep learning (see the book by Goodfellow et al. (2016, Chapter 20) for an excellent overview), and we firstly present *bias-variance decomposition for Boltzmann machines*. The key to achieve this analysis is to employ an *information geometric formulation* of a hierarchical probabilistic model, which was firstly explored by Amari (2001); Nakahara & Amari (2002); Nakahara et al. (2006). In particular, the recent advances of the formulation by Sugiyama et al. (2016; 2017) enables us to analytically obtain the Fisher information of parameters in Boltzmann machines, which is essential to give the lower bound of variances in bias-variance decomposition.

We show an interesting phenomenon revealed by our bias-variance decomposition: The variance does not necessarily increase while the bias always monotonically decreases when we include more parameters in Boltzmann machines, which is caused by its hierarchical structure. Our result indicates the possibility of designing a deep learning architecture that can reduce both of bias and variance, leading to better generalization ability with keeping the representation power.

The remainder of this paper is organized as follows: First we formulate the log-linear model of hierarchical probability distributions using an information geometric formulation in Section 2, which includes the traditional Boltzmann machines (Section 2.2) and arbitrary-order Boltzmann machines (Section 2.3). Then we present the main result of this paper, bias-variance decomposition for Boltzmann machines, in Section 3 and discuss its property. We empirically evaluate the tightness of our theoretical lower bound of the variance in Section 4. Finally, we conclude with summarizing the contribution of this paper in Section 5.

## 2 FORMULATION

To theoretically analyze learning of Boltzmann machines (Ackley et al., 1985), we introduce an information geometric formulation of the log-linear model of hierarchical probability distributions, which can be viewed as a generalization of Boltzmann machines.

## 2.1 Preliminary: Log-Linear Model

First we prepare a log-linear probabilistic model on a partial order structure, which has been introduced by Sugiyama et al. (2016; 2017). Let $(S, \leq)$ be a *partially ordered set*, or a *poset* (Gierz et al., 2003), where a *partial order* $\leq$ is a relation between elements in $S$ satisfying the following three properties for all $x, y, z \in S$: (1) $x \leq x$ (reflexivity), (2) $x \leq y, y \leq x \Rightarrow x = y$ (antisymmetry), and (3) $x \leq y, y \leq z \Rightarrow x \leq z$ (transitivity). We assume that $S$ is always finite and includes the least element (bottom) $\perp \in S$; that is, $\perp \leq x$ for all $x \in S$. We denote $S \setminus \{\perp\}$ by $S^+$.

We use two functions, the *zeta function* $\zeta \colon S \times S \to \{0, 1\}$ and the *Möbius function* $\mu \colon S \times S \to \mathbb{Z}$. The zeta function $\zeta$ is defined as $\zeta(s, x) = 1$ if $s \leq x$ and $\zeta(s, x) = 0$ otherwise, while the Möbius function $\mu$ is its convolutional inverse, that is,

$$\sum_{s \in S} \zeta(x, s) \mu(s, y) = \sum_{x \leq s \leq y} \mu(s, y) = \begin{cases} 1 & \text{if } x = y, \\ 0 & \text{otherwise,} \end{cases}$$

which is inductively defined as

$$\mu(x, y) = \begin{cases} 1 & \text{if } x = y, \\ -\sum_{x \leq s < y} \mu(x, s) & \text{if } x < y, \\ 0 & \text{otherwise.} \end{cases}$$

For any functions $f$, $g$, and $h$ with the domain $S$ such that

$$g(x) = \sum_{s \in S} \zeta(s, x) f(s) = \sum_{s \leq x} f(s), \quad h(x) = \sum_{s \in S} \zeta(x, s) f(s) = \sum_{s \geq x} f(s),$$

$f$ is uniquely recovered using the Möbius function:

$$f(x) = \sum_{s \in S} \mu(s, x) g(s), \quad f(x) = \sum_{s \in S} \mu(x, s) h(s).$$

This is the *Möbius inversion formula* and is fundamental in enumerative combinatorics (Ito, 1993).

Sugiyama et al. (2017) introduced a *log-linear model* on $S$, which gives a discrete probability distribution with the structured outcome space $(S, \leq)$. Let $P$ denote a probability distribution that assigns a probability $p(x)$ for each $x \in S$ satisfying $\sum_{x \in S} p(x) = 1$. Each probability $p(x)$ for $x \in S$ is defined as

$$\log p(x) \coloneqq \sum_{s \in S} \zeta(s, x) \theta(s) = \sum_{s \leq x} \theta(s). \tag{1}$$

From the Möbius inversion formula, $\theta$ is obtained as

$$\theta(x) = \sum_{s \in S} \mu(s, x) \log p(s). \tag{2}$$

In addition, we introduce $\eta \colon S \to \mathbb{R}$ as

$$\eta(x) \coloneqq \sum_{s \in S} \zeta(x, s) p(s) = \sum_{s \geq x} p(s), \quad p(x) = \sum_{s \in S} \mu(x, s) \eta(s). \tag{3}$$

The second equation is from the Möbius inversion formula. Sugiyama et al. (2017) showed that the set of distributions $\boldsymbol{\mathcal{S}} = \{P \mid 0 < p(x) < 1 \text{ and } \sum p(x) = 1\}$ always becomes the *dually flat Riemannian manifold*. This is why two functions $\theta$ and $\eta$ are dual coordinate systems of $\boldsymbol{\mathcal{S}}$ connected with the Legendre transformation, that is,

$$\theta = \nabla \varphi(\eta), \quad \eta = \nabla \psi(\theta)$$

with two convex functions

$$\psi(\theta) \coloneqq -\theta(\perp) = -\log p(\perp), \quad \varphi(\eta) \coloneqq \sum_{x \in S} p(x) \log p(x).$$

Moreover, the Riemannian metric $g(\xi)$ ($\xi = \theta$ or $\eta$) such that

$$g(\theta) = \nabla \nabla \psi(\theta), \quad g(\eta) = \nabla \nabla \varphi(\eta),$$

which corresponds to the gradient of $\theta$ or $\eta$, is given as

$$g_{xy}(\theta) = \frac{\partial \eta(x)}{\partial \theta(y)} = \mathbf{E}\left[\frac{\log p(s)}{\theta(x)}\frac{\log p(s)}{\theta(y)}\right] = \sum_{s \in S} \zeta(x,s)\zeta(y,s)p(s) - \eta(x)\eta(y), \tag{4}$$

$$g_{xy}(\eta) = \frac{\partial \theta(x)}{\partial \eta(y)} = \mathbf{E}\left[\frac{\log p(s)}{\eta(x)}\frac{\log p(s)}{\eta(y)}\right] = \sum_{s \in S} \mu(s,x)\mu(s,y)p(s)^{-1}. \tag{5}$$

for all $x, y \in S^+$. Furthermore, $\boldsymbol{S}$ is in the exponential family (Sugiyama et al., 2016), where $\theta$ coincides with the *natural parameter* and $\eta$ with the *expectation parameter*.

Let us consider two types of submanifolds:

$$\boldsymbol{S}_\alpha = \{\, P \in \boldsymbol{S} \mid \theta(x) = \alpha(x) \text{ for all } x \in \mathrm{dom}(\alpha)\,\},$$
$$\boldsymbol{S}_\beta = \{\, P \in \boldsymbol{S} \mid \eta(x) = \beta(x) \text{ for all } x \in \mathrm{dom}(\beta)\,\}$$

specified by two functions $\alpha, \beta$ with $\mathrm{dom}(\alpha), \mathrm{dom}(\beta) \subseteq S^+$, where the former submanifold $\boldsymbol{S}_\alpha$ has constraints on $\theta$ while the latter $\boldsymbol{S}_\beta$ has those on $\eta$. It is known in information geometry that $\boldsymbol{S}_\alpha$ is $e$-flat and $\boldsymbol{S}_\beta$ is $m$-flat, respectively (Amari, 2016, Chapter 2.4). Suppose that $\mathrm{dom}(\alpha) \cup \mathrm{dom}(\beta) = S^+$ and $\mathrm{dom}(\alpha) \cap \mathrm{dom}(\beta) = \emptyset$. Then the intersection $\boldsymbol{S}_\alpha \cap \boldsymbol{S}_\beta$ is always the singleton, that is, the distribution $Q$ satisfying $Q \in \boldsymbol{S}_\alpha$ and $Q \in \boldsymbol{S}_\beta$ always uniquely exists, and the following *Pythagorean theorem* holds:

$$D_{\mathrm{KL}}(P, R) = D_{\mathrm{KL}}(P, Q) + D_{\mathrm{KL}}(Q, R), \tag{6}$$
$$D_{\mathrm{KL}}(R, P) = D_{\mathrm{KL}}(R, Q) + D_{\mathrm{KL}}(Q, P) \tag{7}$$

for any $P \in \boldsymbol{S}_\alpha$ and $R \in \boldsymbol{S}_\beta$.

## 2.2 Standard Boltzmann Machines

A *Boltzmann machine* is represented as an undirected graph $G = (V, E)$ with a vertex set $V = \{1, 2, \ldots, n\}$ and an edge set $E \subseteq \{\{i, j\} \mid i, j \in V\}$. The energy function $\Phi : \{0, 1\}^n \to \mathbb{R}$ of the Boltzmann machine $G$ is defined as

$$\Phi(\boldsymbol{x}; \boldsymbol{b}, \boldsymbol{w}) = -\sum_{i \in V} b_i x_i - \sum_{\{i,j\} \in E} w_{ij} x_i x_j$$

for each $\boldsymbol{x} = (x_1, x_2, \ldots, x_n) \in \{0, 1\}^n$, where $\boldsymbol{b} = (b_1, b_2, \ldots, b_n)$ and $\boldsymbol{w} = (w_{12}, w_{13}, \ldots, w_{n-1n})$ are parameter vectors for vertices (bias) and edges (weight), respectively, such that $w_{ij} = 0$ if $\{i, j\} \notin E$. The probability $p(\boldsymbol{x}; \boldsymbol{b}, \boldsymbol{w})$ of the Boltzmann machine $G$ is obtained for each $\boldsymbol{x} \in \{0, 1\}^n$ as

$$p(\boldsymbol{x}; \boldsymbol{b}, \boldsymbol{w}) = \frac{\exp(-\Phi(\boldsymbol{x}; \boldsymbol{b}, \boldsymbol{w}))}{Z} \tag{8}$$

with the partition function $Z$ such that

$$Z = \sum_{\boldsymbol{x} \in \{0,1\}^n} \exp(-\Phi(\boldsymbol{x}; \boldsymbol{b}, \boldsymbol{w})) \tag{9}$$

to ensure the condition $\sum_{\boldsymbol{x} \in \{0,1\}^n} p(\boldsymbol{x}; \boldsymbol{b}, \boldsymbol{w}) = 1$.

It is clear that a Boltzmann machine is a special case of the log-linear model in Equation (1) with $S = 2^V$, the power set of $V$, and $\bot = \emptyset$. Let each $x \in S$ be the set of indices of "1" of $\boldsymbol{x} \in \{0, 1\}^n$ and $\leq$ be the inclusion relation, that is, $x \leq y$ if and only if $x \subseteq y$. Suppose that

$$B = \{\, x \in S^+ \mid |x| = 1 \text{ or } x \in E\,\}, \tag{10}$$

where $|x|$ is the cardinality of $x$, which we call a *parameter set*. The Boltzmann distribution in Equations (8) and (9) directly corresponds to the log-linear model in Equation (1):

$$\log p(x) = \sum_{s \in B} \zeta(s, x)\theta(s) - \psi(\theta), \quad \psi(\theta) = -\theta(\bot) = \log Z, \tag{11}$$

where $\theta(x) = b_x$ if $|x| = 1$ and $\theta(x) = w_x$ if $|w| = 2$. This means that the set of Boltzmann distributions $\boldsymbol{S}(B)$ that can be represented by a parameter set $B$ is a *submanifold* of $\boldsymbol{S}$ given as

$$\boldsymbol{S}(B) := \{\, P \in \boldsymbol{S} \mid \theta(x) = 0 \text{ for all } x \notin B \,\}. \tag{12}$$

Given an empirical distribution $\hat{P}$. Learning of a Boltzmann machine is to find the best approximation of $\hat{P}$ from the Boltzmann distributions $\boldsymbol{S}(B)$, which is formulated as a minimization problem of the KL (Kullback–Leibler) divergence:

$$\min_{P_B \in \boldsymbol{S}(B)} D_{\mathrm{KL}}(\hat{P}, P_B) = \min_{P_B \in \boldsymbol{S}(B)} \sum_{s \in S} \hat{p}(s) \log \frac{\hat{p}(s)}{p_B(s)}. \tag{13}$$

This is equivalent to maximize the log-likelihood $L(P_B) = N \sum_{s \in S} \hat{p}(s) \log p_B(s)$ with the sample size $N$. Since we have

$$\frac{\partial}{\partial \theta_B(x)} D_{\mathrm{KL}}(\hat{P}, P_B) = \frac{\partial}{\partial \theta_B(x)} \sum_{s \in S} \hat{p}(s) \log p_B(s)$$

$$= \frac{\partial}{\partial \theta_B(x)} \sum_{s \in S} \left( \hat{p}(s) \sum_{\perp < u \leq s} \theta_B(u) \right) - \frac{\partial}{\partial \theta_B(x)} \psi(\theta_B) \sum_{s \in S} \hat{p}(s)$$

$$= \hat{\eta}(x) - \eta_B(x),$$

the KL divergence $D_{\mathrm{KL}}(\hat{P}, P_B)$ is minimized when $\hat{\eta}(x) = \eta_B(x)$ for all $x \in B$, which is well known as the *learning equation of Boltzmann machines* as $\hat{\eta}(x)$ and $\eta_B(x)$ coincides with the *expectation* for the outcome $x$ with respect to the empirical distribution $\hat{P}$ obtained from data and the model distribution $P_B$ represented by a Boltzmann Machine, respectively. Thus the minimizer $P_B \in \boldsymbol{S}(B)$ of the KL divergence $D_{\mathrm{KL}}(\hat{P}, P_B)$ is the distribution given as

$$\begin{cases} \eta_B(x) = \hat{\eta}(x) & \text{if } x \in B \cup \{\perp\}, \\ \theta_B(x) = 0 & \text{otherwise.} \end{cases} \tag{14}$$

This distribution $P_B$ is known as *m-projection* of $\hat{P}$ onto $\boldsymbol{S}(B)$ (Sugiyama et al., 2017), which is unique and always exists as $\boldsymbol{S}$ has the dually flat structure with respect to $(\theta, \eta)$.

## 2.3 ARBITRARY-ORDER BOLTZMANN MACHINES

The parameter set $B$ is fixed in Equation (10) in the traditional Boltzmann machines, but our log-linear formulation allows us to include or remove any element in $S^+ = 2^V \setminus \{\perp\}$ as a parameter. This attempt was partially studied by Sejnowski (1986); Min et al. (2014) that include higher order interactions of variables to increase the representation power of Boltzmann machines. For $S = 2^V$ with $V = \{1, 2, \ldots, n\}$ and a parameter set $B \subseteq S^+$, which is an arbitrary subset of $S^+ = S \setminus \{\emptyset\}$, the Boltzmann distribution given by an *arbitrary-order Boltzmann machine* is defined by

$$\log p(x) = \sum_{s \in B} \zeta(s, x)\theta(s) - \psi(\theta), \quad \psi(\theta) = -\theta(\perp),$$

and the submanifold of Boltzmann distributions is given by Equation (12). Hence Equation (14) gives the MLE (maximum likelihood estimation) of the empirical distribution $\hat{P}$.

Let $B_1, B_2, \ldots, B_m$ be a sequence of parameter sets such that

$$B_1 \subseteq B_2 \subseteq \cdots \subseteq B_{m-1} \subseteq B_m = S^+.$$

Since we have a hierarchy of submanifolds

$$\boldsymbol{S}(B_1) \subseteq \boldsymbol{S}(B_2) \subseteq \cdots \subseteq \boldsymbol{S}(B_{m-1}) \subseteq \boldsymbol{S}(B_m) = \boldsymbol{S},$$

we obtain the decreasing sequence of KL divergences:

$$D_{\mathrm{KL}}(\hat{P}, P_{B_1}) \geq D_{\mathrm{KL}}(\hat{P}, P_{B_2}) \geq \cdots \geq D_{\mathrm{KL}}(\hat{P}, P_{B_{m-1}}) \geq D_{\mathrm{KL}}(\hat{P}, P_{B_m}) = 0, \tag{15}$$

where each $P_{B_i} = \mathrm{argmin}_{P \in \boldsymbol{S}(B_i)} D_{\mathrm{KL}}(\hat{P}, P)$, the best approximation of $\hat{P}$ using $B_i$.

There are two extreme cases as a choice of the parameter set $B$. On the one hand, if $B = \emptyset$, the Boltzmann distribution is always the uniform distribution, that is, $p(x) = 1/2^{|V|}$ for all $x \in S$. Thus there is no variance but nothing will be learned from data. On the other hand, if $B = S^+$, the Boltzmann distribution can always exactly represent the empirical distribution $\hat{P}$, that is, $D_{\mathrm{KL}}(\hat{P}, P_B) = D_{\mathrm{KL}}(\hat{P}, \hat{P}) = 0$. Thus there is no bias in each training but the variance across different samples will be large. To analyze the tradeoff between the bias and the variance, we perform bias-variance decomposition in the next section.

Another strategy to increase the representation power is to use *hidden variables* (Le Roux & Bengio, 2008) such as restricted Boltzmann machines (RBMs) (Smolensky, 1986; Hinton, 2002) and deep Boltzmann machines (DBMs) (Salakhutdinov & Hinton, 2009; 2012). A Boltzmann machine with hidden variables is represented as $G = (V \cup H, E)$, where $V$ and $H$ correspond to visible and hidden variables, respectively, and the resulting domain $S = 2^{V \cup H}$ (see Appendix for the formulation of Boltzmann machines with hidden variables as the log-linear model). It is known that the resulting model can be singular (Yamazaki & Watanabe, 2005; Watanabe, 2007) and its statistical analysis cannot be directly performed. Studying bias-variance decomposition for such Boltzmann machines with hidden variables is the interesting future work.

## 3 BIAS-VARIANCE DECOMPOSITION

Here we present the main result of this paper, *bias-variance decomposition for Boltzmann machines*. We focus on the expectation of the squared KL divergence $\mathbf{E}[D_{\mathrm{KL}}(P^*, \hat{P}_B)^2]$ from the true (unknown) distribution $P^*$ to the MLE $\hat{P}_B$ of an empirical distribution $\hat{P}$ by a Boltzmann machine with a parameter set $B$, and decompose it using information geometric properties.

In the following, we use the MLE $P_B^*$ of the true distribution $P^*$, which is the closest distribution in the set of distributions that can be modeled by Boltzmann machines in terms of the KL divergence and is mathematically obtained with replacing $\hat{P}$ with $P^*$ in Equation (13).

**Theorem 1** (Bias-variance decomposition of the KL divergence)**.** *Given a Boltzmann machine with a parameter set $B$. Let $P^* \in \mathcal{S}$ be the true (unknown) distribution, $P_B^*, \hat{P}_B \in \mathcal{S}(B)$ be the MLEs of $P^*$ and an empirical distribution $\hat{P}$, respectively. We have*

$$\mathbf{E}\Big[D_{\mathrm{KL}}(P^*, \hat{P}_B)^2\Big] = D_{\mathrm{KL}}(P^*, P_B^*)^2 + \mathbf{E}\Big[D_{\mathrm{KL}}(P_B^*, \hat{P}_B)^2\Big]$$

$$= \underbrace{D_{\mathrm{KL}}(P^*, P_B^*)^2}_{\text{bias}^2} + \underbrace{\mathrm{var}(P_B^*, B)}_{\text{variance}} + \text{irreducible error}$$

$$\geq D_{\mathrm{KL}}(P^*, P_B^*)^2 + \underline{\mathrm{var}}(P_B^*, B) + \text{irreducible error},$$

$$\mathrm{var}(P_B^*, B) = \sum_{s \in B} \sum_{u \in B} \eta^*(s)\eta^*(u)\mathrm{cov}\Big(\hat{\theta}_B(s), \hat{\theta}_B(u)\Big),$$

$$\underline{\mathrm{var}}(P_B^*, B) = \frac{1}{N} \sum_{s \in B} \sum_{u \in B} \eta^*(s)\eta^*(u)(I^{-1})_{su}$$

*with the equality holding when the sample size $N \to \infty$, where $\mathrm{cov}(\hat{\theta}_B(s), \hat{\theta}_B(u))$ denotes the error covariance between $\hat{\theta}_B(s)$ and $\hat{\theta}_B(u)$ and $I^{-1}$ is the inverse of the Fisher information submatrix $I \in \mathbb{R}^{|B| \times |B|}$ of $P_B^*$ with respect to the parameter set $B$ given in Equation (4).*

*Proof.* From the Pythagorean theorem illustrated in Figure 1,

$$\mathbf{E}\Big[D_{\mathrm{KL}}(P^*, \hat{P}_B)^2\Big] = \mathbf{E}\Big[\big(D_{\mathrm{KL}}(P^*, P_B^*) + D_{\mathrm{KL}}(P_B^*, \hat{P}_B)\big)^2\Big]$$

$$= \mathbf{E}\Big[D_{\mathrm{KL}}(P^*, P_B^*)^2 + 2D_{\mathrm{KL}}(P^*, P_B^*)D_{\mathrm{KL}}(P_B^*, \hat{P}_B) + D_{\mathrm{KL}}(P_B^*, \hat{P}_B)^2\Big]$$

$$= D_{\mathrm{KL}}(P^*, P_B^*)^2 + 2D_{\mathrm{KL}}(P^*, P_B^*)\mathbf{E}\Big[D_{\mathrm{KL}}(P_B^*, \hat{P}_B)\Big] + \mathbf{E}\Big[D_{\mathrm{KL}}(P_B^*, \hat{P}_B)^2\Big].$$

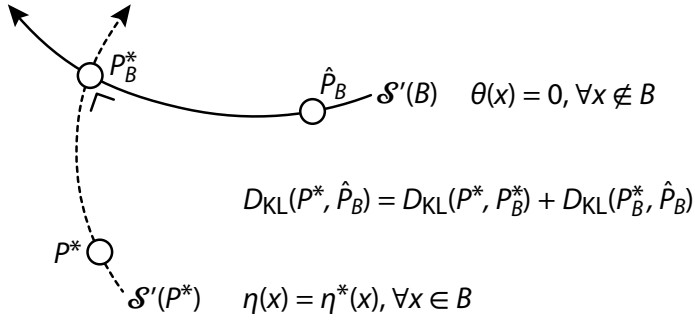

Figure 1: Illustration of the Pythagorean theorem.

Since $\hat{\theta}_B(s)$ is an unbiased estimator of $\theta_B^*(s)$ for every $s \in S$, it holds that

$$\mathbf{E}\big[D_{\mathrm{KL}}(P_B^*, \hat{P}_B)\big] = \mathbf{E}\left[\sum_{x \in S} p_B^*(x) \log \frac{p_B^*(x)}{\hat{p}_B(x)}\right] = \sum_{x \in S} p_B^*(x) \sum_{s \leq x} \mathbf{E}\left[\theta_B^*(s) - \hat{\theta}_B(s)\right] = 0.$$

Hence we have

$$\mathbf{E}\left[D_{\mathrm{KL}}(P^*, \hat{P}_B)^2\right] = D_{\mathrm{KL}}(P^*, P_B^*)^2 + \mathbf{E}\left[D_{\mathrm{KL}}(P_B^*, \hat{P}_B)^2\right]. \tag{16}$$

The second term is

$$\mathbf{E}\left[D_{\mathrm{KL}}(P_B^*, \hat{P}_B)^2\right]$$

$$= \mathbf{E}\left[\left(\sum_{x \in S} p_B^*(x) \log \frac{p_B^*(x)}{\hat{p}_B(x)}\right)^2\right] = \mathbf{E}\left[\sum_{x \in S} \sum_{y \in S} p_B^*(x) p_B^*(y) \log \frac{p_B^*(x)}{\hat{p}_B(x)} \log \frac{p_B^*(y)}{\hat{p}_B(y)}\right]$$

$$= \mathbf{E}\left[\sum_{x \in S} \sum_{y \in S} p_B^*(x) p_B^*(y) \left(\sum_{s \in B, s \leq x} \Big(\theta_B^*(s) - \hat{\theta}_B(s)\Big) - \Big(\psi(\theta_B^*) - \psi(\hat{\theta}_B)\Big)\right)\right.$$

$$\left.\left(\sum_{u \in B, u \leq y} \Big(\theta_B^*(u) - \hat{\theta}_B(u)\Big) - \Big(\psi(\theta_B^*) - \psi(\hat{\theta}_B)\Big)\right)\right]$$

$$= \mathbf{E}\left[\sum_{x \in S} \sum_{y \in S} \sum_{s \in B} \sum_{u \in B} p_B^*(x) p_B^*(y) \zeta(s,x) \zeta(u,y) \Big(\theta_B^*(s) - \hat{\theta}_B(s)\Big) \Big(\theta_B^*(u) - \hat{\theta}_B(u)\Big)\right]$$

$$-2 \mathbf{E}\left[\sum_{x \in S} p_B^*(x) \sum_{s \in B, s \leq x} \Big(\theta_B^*(s) - \hat{\theta}_B(s)\Big) \Big(\psi(\theta_B^*) - \psi(\hat{\theta}_B)\Big)\right] + \mathbf{E}\left[\Big(\psi(\theta_B^*) - \psi(\hat{\theta}_B)\Big)^2\right]$$

$$= \sum_{s \in B} \sum_{u \in B} \eta_B^*(s) \eta_B^*(u) \mathrm{cov}\Big(\hat{\theta}_B(s), \hat{\theta}_B(u)\Big) + \mathrm{var}\Big(\psi(\hat{\theta}_B)\Big) - 2 \sum_{s \in B} \eta_B^*(s) \mathrm{cov}\Big(\hat{\theta}_B(s), \psi(\hat{\theta}_B)\Big)$$

$$= \sum_{s \in B} \sum_{u \in B} \eta^*(s) \eta^*(u) \mathrm{cov}\Big(\hat{\theta}_B(s), \hat{\theta}_B(u)\Big) + \mathrm{var}\Big(\psi(\hat{\theta}_B)\Big) - 2 \sum_{s \in B} \eta^*(s) \mathrm{cov}\Big(\hat{\theta}_B(s), \psi(\hat{\theta}_B)\Big),$$

where $\mathrm{cov}(\hat{\theta}_B(s), \hat{\theta}_B(u))$ denotes the error covariance between $\hat{\theta}_B(s)$ and $\hat{\theta}_B(u)$ and $\mathrm{var}(\psi(\hat{\theta}_B))$ denotes the variance of $\psi(\hat{\theta}_B)$, and the last equality comes from the condition in Equation (14). Here the term of the (co)variance of the normalizing constant (partition function) $\psi(\theta)$:

$$\mathrm{var}\Big(\psi(\hat{\theta}_B)\Big) - 2 \sum_{s \in B} \eta^*(s) \mathrm{cov}\Big(\hat{\theta}_B(s), \psi(\hat{\theta}_B)\Big)$$

is the irreducible error since $\psi(\theta) = -\theta(\perp)$ is orthogonal for every parameter $\theta(s)$, $s \in S$ and the Fisher information vanishes from Equation (4), i.e.,

$$\mathbf{E}\left[\frac{\partial \log p(s)}{\partial \theta(s)} \frac{\partial \log p(s)}{\partial \theta(\perp)}\right] = \sum_{s \in S} \zeta(x,s) p(s) - \eta(x) = \eta(x) - \eta(x) = 0.$$

For the variance term

$$\text{var}(P_B^*, B) = \sum_{s \in B} \sum_{u \in B} \eta^*(s)\eta^*(u)\text{cov}\left(\hat{\theta}_B(s), \hat{\theta}_B(u)\right),$$

we have from the Cramér–Rao bound (Amari, 2016, Theorem 7.1) since $\theta_B(s)$, $s \in B$ is unbiased

$$\text{cov}\left(\hat{\theta}_B(s), \hat{\theta}_B(u)\right) \geq \frac{1}{N}(I^{-1})_{su}$$

with the equality holding when $N \to \infty$, where $I \in \mathbb{R}^{|B| \times |B|}$ is the Fisher information matrix with respect to the parameter set $B$ such that

$$I_{su} = \sum_{x \in S} \zeta(s, x)\zeta(u, x)p_B^*(x) - \eta_B^*(s)\eta_B^*(u)$$

for all $s, u \in B$, which is given in Equation (5), and $I^{-1}$ is its inverse. Finally, from Equation (16) we obtain

$$\mathbf{E}\left[D_{\text{KL}}(P^*, \hat{P}_B)^2\right]$$

$$= D_{\text{KL}}(P^*, P_B^*)^2 + \mathbf{E}\left[D_{\text{KL}}(P_B^*, \hat{P}_B)^2\right]$$

$$= D_{\text{KL}}(P^*, P_B^*)^2 + \sum_{s \in B} \sum_{u \in B} \eta^*(s)\eta^*(u)\text{cov}\left(\hat{\theta}_B(s), \hat{\theta}_B(u)\right) + \text{irreducible error}$$

$$\geq D_{\text{KL}}(P^*, P_B^*)^2 + \frac{1}{N} \sum_{s \in B} \sum_{u \in B} \eta^*(s)\eta^*(u)(I^{-1})_{su} + \text{irreducible error}$$

with the equality holding when $N \to \infty$. $\qquad \square$

Let $B, B' \subseteq S' = 2^{V \cup H}$ such that $B \subseteq B'$, that is, $B'$ has more parameters than $B$. Then it is clear that the bias always reduces, that is,

$$D_{\text{KL}}(P^*, P_B^*) \geq D_{\text{KL}}(P^*, P_{B'}^*)$$

because $\mathcal{S}(B) \subseteq \mathcal{S}(B')$. However, this monotonicity does not always hold for the variance. We illustrate this non-monotonicity in the following example. Let $S = 2^V$ with $V = \{1, 2, 3\}$ and assume that the true distribution $P^*$ is given by

$$\left(p^*(\{\emptyset\}), p^*(\{1\}), p^*(\{2\}), p^*(\{3\}), p^*(\{1,2\}), p^*(\{1,3\}), p^*(\{2,3\}), p^*(\{1,2,3\})\right)$$
$$= (0.2144, 0.0411, 0.2037, 0.145, 0.1423, 0.0337, 0.0535, 0.1663),$$

which was randomly generated from the uniform distribution. Suppose that we have three types of parameter sets $B_1 = \{\{1\}\}$, $B_2 = \{\{1\}, \{2\}\}$, and $B_3 = \{\{1\}, \{2\}, \{1,2\}\}$ such that $B_1 \subset B_2 \subset B_3$. Then biases become

$$D_{\text{KL}}(P^*, P_{B_1}^*)^2 = 0.0195, \quad D_{\text{KL}}(P^*, P_{B_2}^*)^2 = 0.0172, \quad D_{\text{KL}}(P^*, P_{B_3}^*)^2 = 0.0030,$$

where we can confirm that the bias monotonically decreases, while the lower bound of the variance with $N = 100$ are

$$\underline{\text{var}}(P_{B_1}^*, B_1) = 0.0062, \quad \underline{\text{var}}(P_{B_2}^*, B_2) = 0.0192, \quad \underline{\text{var}}(P_{B_3}^*, B_3) = 0.0178.$$

Moreover, we have computed the exact variance $\text{var}(P_B^*, B)$ by repeating $1,000$ times generating a sample with $N = 100$ and obtained the following:

$$\text{var}(P_{B_1}^*, B_1) = 0.0066, \quad \text{var}(P_{B_2}^*, B_2) = 0.0267, \quad \text{var}(P_{B_3}^*, B_3) = 0.0190,$$

hence $\text{var}(P_{B_2}^*, B_2) > \text{var}(P_{B_3}^*, B_3)$ happens, that is, the variance actually decreases when we include more parameters. This interesting property, *non-monotonicity of the variance with respect to the growth of parameter sets*, comes from the hierarchical structure of $S$.

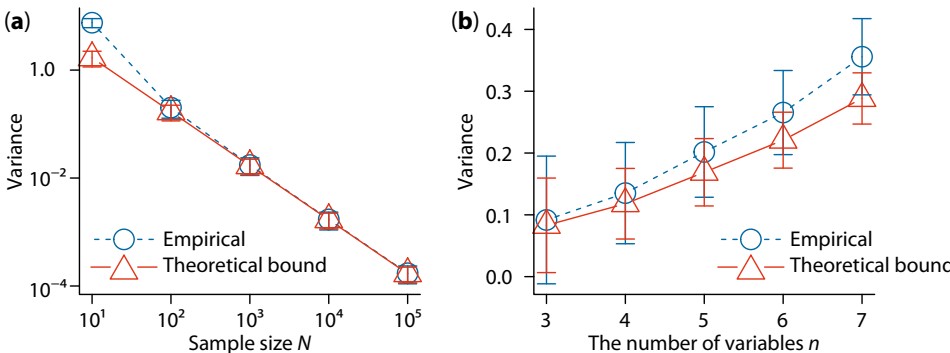

Figure 2: Empirical evaluation of variance. Empirically estimated variances (blue, dotted lines) and theoretically obtained lower bounds (red, solid line) with fixing $n = 5$ in (**a**) and $N = 100$ in (**b**).

## 4 EMPIRICAL EVALUATION OF TIGHTNESS

We empirically evaluate the tightness of our lower bound. In each experiment, we randomly generated a true distribution $P^*$, followed by repeating $1,000$ times generating a sample (training data) with the size $N$ from $P^*$ to empirically estimate the variance $\mathrm{var}(P^*_B, B)$. We consistently used the Boltzmann machine represented as the fully connected graph $G = (V, E)$ such that $V = \{1, 2, \dots, n\}$ and $E = \{\{i, j\} \mid i, j \in V\}$. Thus the parameter set $B$ is given as

$$B = \left\{ x \in S^+ = 2^V \setminus \{\emptyset\} \mid |x| \leq 2 \right\}.$$

We report in Figure 2 the mean $\pm$ SD (standard deviation) of the empirically obtained variance and its theoretical lower bound $\underline{\mathrm{var}}(P^*_B, B)$ obtained by repeating the above estimation 100 times. In Figure 2(**a**) the sample size $N$ is varied from 10 to $10,000$ with fixing the number of variables $n = 5$ while in Figure 2(**b**) $n$ is varied from 3 to 7 with fixing $N = 100$. These results overall show that our theoretical lower bound is tight enough if $N$ is large and is reasonable across each $n$.

## 5 CONCLUSION

In this paper, we have firstly achieved bias-variance decomposition of the KL divergence for Boltzmann machines using the information geometric formulation of hierarchical probability distributions. Our model is a generalization of the traditional Boltzmann machines, which can incorporate arbitrary order interactions of variables. Our bias-variance decomposition reveals the non-monotonicity of the variance with respect to growth of parameter sets, which has been also reported elsewhere for non-linear models (Faber, 1999). This result indicates that it is possible to reduce both bias and variance when we include more higher-order parameters in the hierarchical deep learning architectures. To solve the open problem of the generalizability of the deep learning architectures, our finding can be fundamental for further theoretical development.

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

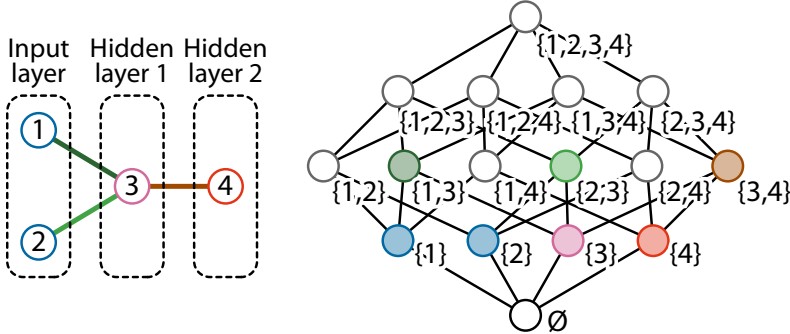

Figure 3: An example of a deep Boltzmann machine (left) with an input (visible) layer $V = \{1, 2\}$ with two hidden layers $H_1 = \{3\}$ and $H_2 = \{4\}$, and the corresponding domain set $S^{V \cup H}$ (right). In the right-hand side, the colored objects $\{1\}, \{2\}, \{3\}, \{4\}, \{1, 3\}, \{2, 3\},$ and $\{3, 4\}$ denote the parameter set $B$, which correspond to nodes and edges of the DBM in the left-hand side.

## A  BOLTZMANN MACHINES WITH HIDDEN NODES

A Boltzmann machine with hidden variables is represented as $G = (V \cup H, E)$, where $V$ and $H$ correspond to visible and hidden variables, respectively, and the resulting domain $S = 2^{V \cup H}$. In particular, restricted Boltzmann machines (RBMs) (Smolensky, 1986; Hinton, 2002) are often used in applications, where the edge set is given as

$$E = \{ \{i, j\} \mid i \in V, j \in H \}$$

Moreover, in a deep Boltzmann machine (DBM) (Salakhutdinov & Hinton, 2009; 2012), which is the beginning of the recent trend of deep learning (LeCun et al., 2015; Goodfellow et al., 2016), the hidden variables $H$ are divided into $k$ disjoint subsets (layers) $H_1, H_2, \ldots, H_k$ and

$$E = \{ \{i, j\} \mid i \in H_{l-1}, j \in H_l, l \in \{1, \ldots, k\} \},$$

where $V = H_0$ for simplicity.

Let $S = 2^V$ and $S' = 2^{V \cup H}$ and $\boldsymbol{S}$ and $\boldsymbol{S}'$ be the set of distributions with the domains $S$ and $S'$, respectively. In both cases of RBMs and DBMs, we have

$$B = \{ x \in S' \mid |x| = 1 \text{ or } x \in E \},$$

(see Figure 3) and the set of Boltzmann distributions is obtained as

$$\boldsymbol{S}'(B) = \{ P \in \boldsymbol{S}' \mid \theta(x) = 0 \text{ for all } x \notin B \}.$$

Since the objective of learning Boltzmann machines with hidden variables is MLE (maximum likelihood estimation) with respect to the marginal probabilities of the visible part, the target empirical distribution $\hat{P} \in \boldsymbol{S}$ is extended to the submanifold $\boldsymbol{S}'(\hat{P})$ such that

$$\boldsymbol{S}'(\hat{P}) = \{ P \in \boldsymbol{S}' \mid \eta(x) = \hat{\eta}(x) \text{ for all } x \in S \},$$

and the process of learning Boltzmann machines with hidden variables is formulated as double minimization of the KL divergence such that

$$\min_{P \in \boldsymbol{S}'(\hat{P}), P_B \in \boldsymbol{S}'(B)} D_{\mathrm{KL}}(P, P_B). \tag{17}$$

Since two submanifolds $\boldsymbol{S}'(B)$ and $\boldsymbol{S}'(\hat{P})$ are $e$-flat and $m$-flat, respectively, it is known that the EM-algorithm can obtain a local optimum of Equation (17) (Amari, 2016, Section 8.1.3), which was first analyzed by Amari et al. (1992). Since this computation is infeasible due to combinatorial explosion of the domain $S' = 2^{V \cup H}$, a number of approximation methods such as Gibbs sampling have been proposed (Salakhutdinov & Hinton, 2012).

Let us fix an empirical distribution $\hat{P}$ and consider two RBMs with parameter sets $B, B' \subseteq S' = 2^{V \cup H}$. If $B \subseteq B'$, that is, $B'$ has more hidden nodes than $B$, we always have the monotonicity:

$$\min_{P \in \boldsymbol{S}'(\hat{P}), P_{B'} \in \boldsymbol{S}'(B')} D_{\mathrm{KL}}(P, P_{B'}) \leq \min_{P \in \boldsymbol{S}'(\hat{P}), P_B \in \boldsymbol{S}'(B)} D_{\mathrm{KL}}(P, P_B) \tag{18}$$

as $B \subseteq B'$ implies $\boldsymbol{S}'(B) \subseteq \boldsymbol{S}'(B')$. This result corresponds to Theorem 1 in (Le Roux & Bengio, 2008), the representation power of RBMs.

