# OpenReview forum: "Bias-Variance Decomposition for Boltzmann Machines"
_ICLR.cc/2018/Conference — Reject_

### Official Review · AnonReviewer1 · 2017-11-24
**The paper presents a interesting analysis revealing the usefulness of analysing the generation ability of ML models based on insights from information geometry. However,  a lower bound on the KL-divergence is less informative in practice.**

**Rating:** 5
**Confidence:** 2

**Review:**

Summary of the paper:
The paper derives a lower bound on the expected  squared KL-divergence between a true distribution and the sample based maximum likelihood estimate (MLE) of that distribution modelled by an Boltzmann machine (BM) based on methods from information geometry. This  KL-divergence is first split into the squared KL-divergence between the true distribution and MLE of that distribution,  and the expected squared KL-divergence between the MLE of the true distribution and the sample based MLE (in a similar spirit to splitting the excess error into approximation and estimation error in statistical learning theory). The letter is than lower bounded (leading to a lower bound on the overall KL-divergence) by a term  which does not necessarily increase if the number of model parameters is increased.


Pros:
- Using insights from information geometry  opens up a very interesting and (to my knowledge) new approach for analysing the generalisation ability of ML models.
- I am not an expert on information geometry and I did not find the time to follow all the steps of the proof in detail, but the analysis seems to be correct.

Cons:
- The fact that the lower bound does not necessary increase with a growing number of parameters does not guarantee that the same holds true for the KL-divergence (in this sense an upper bound would be more informative). Therefore, it is not clear how much of insights the theoretical analysis gives for practitioners (it could be nice to analyse the tightness of the bound for toy models).
- Another drawback reading the practical impact is, that the theorem bounds the expected  squared KL-divergence between a true distribution and the sample based MLE, while training minimises the divergence between the empirical distribution and the model distribution ( i.e. the sample based MLE in the optimal case),  and the theorem does not show the dependency on the letter.

I found some parts difficulty to understand and clarity could be improved  e.g. by
- explaining why minimising KL(\hat P, P_B) is equivalent to minimising the KL-divergence between the empirical distribution and the Gibbs distribution \Phi.
- explaining in which sense the formula on page 4 is equivalent to “the learning equation of Boltzmann machines”.
- explaining what is the MLE of the true distribution (I assume the closest distribution in the set of distributions that can be modelled by the BM).

Minor comments:
- page 1: and DBMs….(Hinton et al., 2006) : The paper describes deep belief networks (DBNs) not DBMs
- \theta is used to describe the function in eq. (2) as well as the BM parameters in Section 2.2
- page 5: “nodes H is” -> “nodes H are”



REVISION:
Thanks to the reviewers for replying to my comments and making the changes. I think they improved the paper. On the other hand the other reviewers raised valid questions, that led to my decision to not change the overall rating of the paper.

---

> ### Author Response · Authors · 2017-12-27
> **Thank you for the review**
>
> Thank you for your valuable comments. We have revised our manuscript according to reviewers’ comments and corrected mistakes. In particular, the bound provided in Theorem 1 has been revised and the example provided Theorem 1 has been updated. Moreover, we have newly added empirical evaluation of our theoretical result. In the following, we answer each question.
>
> > REVIEWER 1: it is not clear how much of insights the theoretical analysis gives for practitioners (it could be nice to analyse the tightness of the bound for toy models).
>
> > ANSWER: We have additionally conducted empirical evaluation of the tightness of our theoretical lower bound in the revised version. Please check Section 4 and Figure 2. We confirm that our lower bound is quite tight in practice.
>
> > REVIEWER 1: the theorem bounds the expected  squared KL-divergence between a true distribution and the sample based MLE, while training minimises the divergence between the empirical distribution and the model distribution, and the theorem does not show the dependency on the letter.
>
> > ANSWER: The KL-divergence between the empirical distribution and the model distribution in each training monotonically decreases if we include more parameters (see Equation (15)). But overfitting surely occurs if we include too many parameters and this is our motivation of performing bias-variance decomposition to analyze the generalizability of BMs. We have added this discussion in the first paragraph in P.5.
>
> > REVIEWER 1: explaining why minimising KL(\hat P, P_B) is equivalent to minimising the KL-divergence between the empirical distribution and the Gibbs distribution \Phi.
>
> > ANSWER: This is because \hat P is the empirical distribution and P_B coincides with the Gibbs distribution \Phi.
>
> > REVIEWER 1: explaining in which sense the formula on page 4 is equivalent to “the learning equation of Boltzmann machines”.
>
> > ANSWER: This is because \hat{\eta}(x) and \eta_B(x) coincide with the expectation for the outcome x with respect to the empirical distribution obtained from data and the model distribution represented by the Boltzmann Machine B, respectively. We have revised the text to clarify this point.
>
> > REVIEWER 1: explaining what is the MLE of the true distribution (I assume the closest distribution in the set of distributions that can be modelled by the BM).
>
> > ANSWER: You are right. The MLE of the true distribution is the closest distribution in the set of distributions that can be modelled by the BM in terms of the KL divergence. We have revised the text to clarify this point.
>
> > REVIEWER 1: page 1: and DBMs….(Hinton et al., 2006) : The paper describes deep belief networks (DBNs) not DBMs
>
> > ANSWER: We have removed this citation and replaced with [Goodfellow et al. (2016, Chapter 20)].
>
> > REVIEWER 1: \theta is used to describe the function in eq. (2) as well as the BM parameters in Section 2.2
>
> > ANSWER: We have changed the symbol in Eq.(2) for consistency.
>
> > REVIEWER 1: page 5: “nodes H is” -> “nodes H are”
>
> > ANSWER: We have corrected this.

---

### Official Review · AnonReviewer2 · 2017-11-27
**This paper uses an information geometric view on hierarchical models to discuss bias a variance in Boltzmann machines, presenting interesting conclusions, whereby some care seems to be needed in the derivations and discussion.**

**Rating:** 5
**Confidence:** 5

**Review:**

This paper uses an information geometric view on hierarchical models to discuss a bias - variance decomposition in Boltzmann machines, presenting interesting conclusions, whereby some more care appears to be needed for making these claims.

The paper arrives at the main conclusion that it is possible to reduce both the bias and the variance in a hierarchical model. The discussion is not specific to deep learning nor to Boltzmann machines, but actually addresses hierarchical exponential family models. The methods pertaining hierarchical models are interesting and presented in a clear way. My concern are the following points:

The main theorem presents only a lower bound, meaning that it provides no guarantee that the variance can indeed be reduced.

The paper seems to ignore that a model with hidden variables may be singular, in which case the Fisher metric is not positive definite and the Cramer Rao bound has no meaning. This interferes with the claims and derivations made in the paper in the case of models with hidden variables. The problem seems to lie in the fact that the presented derivations assume that an optimal distribution in the data manifold is given (see Theorem 1 and proof), effectively making this a discussion about a fully observed hierarchical model. In particular, it is not further specified how to obtain θˆB(s) in page 6 before (13).

Also, in page 5 the paper states that ``it is known that the EM-algorithm can obtain the global optimum of Equation (12) (Amari, 2016, Section 8.1.3)''. However, what is shown in that reference is only that:  (Theorem 8.2., Amari, 2016) ``The KL-divergence decreases monotonically by repeating the E-step and the M-step. Hence, the algorithm converges to an equilibrium.'' A model with hidden variables can have several global and local optimisers (see, e.g. https://arxiv.org/abs/1709.05276). The critical points of the EM algorithm can have a non trivial structure, as has been observed in the case of non negative rank matrix varieties (see, e.g., https://arxiv.org/pdf/1312.5634.pdf).

OTHER

In page 3, ``S_\beta is e-flat and S_\alpha ... '', should this not be the other way around? (See also page 5 last paragraph of Section 2.) Please also indicate the precise location in the provided reference.

All pages up to page 5 are introduction. Section 2.3. as presented is very vague and does not add much to the discussion.

In page 7, please explain E ψ(θˆ )^2 −ψ(θ∗ )^2=0

---

> ### Author Response · Authors · 2017-12-27
> **Thank you for the review**
>
> Thank you for your valuable comments. We have revised our manuscript according to reviewers’ comments and corrected mistakes. In particular, the bound provided in Theorem 1 has been revised and the example provided after Theorem 1 has been updated. Moreover, we have newly added empirical evaluation of our theoretical result. In the following, we answer each question.
>
> > REVIEWER 2: The main theorem presents only a lower bound, meaning that it provides no guarantee that the variance can indeed be reduced.
>
> > ANSWER: We have additionally conducted empirical evaluation of our theoretical lower bound in the revised version. Please check Section 4 and Figure 2. We confirm that our lower bound is quite tight in practice when the sample size N becomes large and the variance reduction actually happens.
>
> > REVIEWER 2: The paper seems to ignore that a model with hidden variables may be singular.
>
> > ANSWER: Thank you very much for pointing this out. You are right and our theoretical results cannot be directly applied to models with hidden variables. Thus we have removed models with hidden variables from our paper and newly added discussion about this issue in the last paragraph in Section 2.3. Please note that our main theoretical contribution is still fully valid.
>
> > REVIEWER 2: A model with hidden variables can have several global and local optimisers. In particular, it is not further specified how to obtain θˆB(s) in page 6 before (13).
>
> > ANSWER: Thank you very much for pointing this out. You are right and we have revised our text (now in Appendix as we have removed the section of models with hidden variables).
>
> > REVIEWER 2: In page 3, ``S_\beta is e-flat and S_\alpha ... '', should this not be the other way around? (See also page 5 last paragraph of Section 2.) Please also indicate the precise location in the provided reference.
>
> > ANSWER: You are right. This should be the other way around. We have corrected this in the revised version and also clarified the location of the reference (Appendix, after Eq.(17)).
>
> > REVIEWER 2: All pages up to page 5 are introduction. Section 2.3. as presented is very vague and does not add much to the discussion.
>
> > ANSWER: Thank you for pointing this out. We have revised and extended Section 2.3. Although Section 2.1 is preliminary, the other parts of Section 2 are not introduction but necessary discussion to formulate the family of Boltzmann machines as the log-linear model.
>
> > REVIEWER 2: In page 7, please explain E ψ(θˆ )^2 −ψ(θ∗ )^2=0
>
> > ANSWER: Thank you for pointing this out. This was wrong and now corrected. This is indeed irreducible error as the Fisher information vanishes. Please check the revised Theorem 1 and its proof.

---

### Official Review · AnonReviewer3 · 2017-11-28
**Bias-Variance Decomposition for Boltzmann Machines Review**

**Rating:** 7
**Confidence:** 5

**Review:**

Summary: The goal of this paper is to analyze the effectiveness and generalizability of deep learning. This authors present a theoretical analysis of bias-variance decomposition for hierarchical graphical models, specifically Boltzmann Machines (BM).  The analysis follows a geometric formulation of hierarchical probability distributions. The authors describe a general log-linear model and other variations of it such as the standard BM, arbitrary-order BM and Restricted BM to motivate their approach.

The authors first define the bias-variance decomposition of KL divergence using Pythagorean theorem followed by applying Cramer-Rao bound and show that the variance decreases when adding more parameters in the model.

Positives:
-The paper is clearly written and the analysis is helpful to show the effect of adding more parameters on the variance and bias in a general architecture (the Boltzmann Machines)
-The authors did a good job covering general probabilistic models and progression of models starting with the log-linear model.
-The authors provided an example to illustrate the theory, by showing that the variance decreases with the increase of model parameters.

Questions:
-How does this analysis apply to other deep learning architectures such as Convolutional Neural Networks?
-How does this analysis apply to other frameworks such as variational auto-encoders and generative adversarial networks?

---

> ### Author Response · Authors · 2017-12-27
> **Thank you for the review**
>
> Thank you for your valuable comments. We have revised our manuscript according to reviewers’ comments and corrected mistakes. In particular, the bound provided in Theorem 1 has been revised and the example provided after Theorem 1 has been updated. Moreover, we have newly added empirical evaluation of our theoretical result.
> Analyzing the relationship between our model and such neural network models suggested in you comments, in particular probabilistic models of variational auto-encoders and generative adversarial networks, is not in the scope of this paper but our exciting future topic.

---

### Decision · Program_Chairs · 2018-01-29
**ICLR 2018 Conference Acceptance Decision**

**Decision:**

Reject

**Comment:**

This paper presents a bias/variance decomposition for Boltzmann machines using the generalized Pythagorean Theorem from information geometry. The main conclusion is that counterintuitively, the variance may decrease as the model is made larger. There are probably some interesting ideas here, but there isn't a clear take-away message, and it's not clear how far this goes beyond previous work on estimation of exponential families (which is a well-studied topic).

Some of the reviewers caught mathematical errors in the original draft; the revised version fixed these, but did so partly by removing a substantial part of the paper about hidden variables. The analysis, then, is limited to fully observed Boltzmann machines, which have less practical interest to the field of deep learning.